# Molecular and Technological Characterization of *Saccharomyces cerevisiae* from Sourdough

Furkan Aydın [1],*, Tacettin Utku Günen [2], Halil İbrahim Kahve [1], Emrah Güler [3], Göksel Özer [2], Yeşim Aktepe [4] and İbrahim Çakır [4]

1 Department of Food Engineering, Faculty of Engineering, Aksaray University, Aksaray 68100, Turkey
2 Department of Plant Protection, Faculty of Agriculture, Bolu Abant Izzet Baysal University, Bolu 14030, Turkey
3 Department of Horticulture, Faculty of Agriculture, Bolu Abant Izzet Baysal University, Bolu 14030, Turkey
4 Department of Food Engineering, Faculty of Engineering, Bolu Abant Izzet Baysal University, Bolu 14030, Turkey
* Correspondence: furkanaydin06@gmail.com; Tel.: +90-530-692-2917

**Abstract:** DNA markers help detect the intraspecific genetic diversity of yeast strains. Eight ISSR (Inter Simple Sequence Repeats) primers were used to assess the intraspecific diversity of *Saccharomyces cerevisiae* ($n = 96$) from different populations ($n = 3$), evaluate the technological characteristics, and investigate trait-loci associations. The primers amplified 154 reproducible and scorable bands, of which 79.87% were polymorphic. The UPGMA (unweighted pair group method with arithmetic mean) dendrogram clustered 96 isolates into two main clusters, supported by STRUCTURE HARVESTER results ($\Delta K = 2$). Analysis of molecular variance (AMOVA) indicated significant genetic differences between (15%) and within the populations (85%) ($p < 0.001$). Twenty-nine genetically distinct strains were selected for the technological characterization. Principal component analysis (PCA) revealed that five strains with high fermentation capacity, leavening activity, high growth index at 37 °C, and harsh growth conditions were technologically relevant. Trait-loci association analyses indicated that the highest correlation ($r = 0.60$) was recorded for the fermentation capacity on the 8th and 113th loci, amplified by ISSR-1 and ISSR-6 primers, respectively ($p < 0.05$). The strains yielding high performances and the associated loci amplified by ISSR markers possess a high potential to generate locus-specific primers to target the strains with high fermentation capacity.

**Keywords:** sourdough; yeast; *Saccharomyces cerevisiae*; DNA markers; ISSR; genetic variation

## 1. Introduction

Traditional sourdough, a flour-and-water mixture fermented by endogenous lactic acid bacteria (LAB) and yeast, has been used for thousands of years to produce cereal-based fermented goods worldwide [1]. It is a massive source of various endogenous LAB and yeast species and strains due to their artisanal and location-specific handling [2]. The yeasts in sourdough must endure the unique and demanding microbial environment characterized by low pH and oxygen tension and the need to share carbohydrates with the competing LAB communities [3]. *Saccharomyces cerevisiae* has been reported as the predominant yeast species in many sourdough ecosystems [4–8]. It can quickly produce $CO_2$ from sugar, resulting in dough expansion during the fermentation. It significantly influences the texture and flavor development by secreting specific compounds, such as glutathione, glycerol, alcohols, aldehydes, acetoin, and esters [9–11]. These characteristics are strain-specific and mainly dependent on heterogeneity [12].

Molecular markers are widely used for the classification and genetic characterization of yeast species, providing beneficial information in revealing heterogeneity between the strains by producing polymorphic DNA fingerprints and clustering genetically diverse strains separately [13]. In contrast to many other statistical methods, cluster analyses

offer genetically distinct groups and may be used when the research is in the exploratory stage, and when prior assumptions are lacking [14]. Thus, reducing the number of isolates that should be technologically analyzed is possible by choosing strains from the various clusters. Additionally, the association between polymorphic loci and technological traits may make it possible for researchers to use the sequence-characterized amplified regions (SCAR) approach to target more specific gene regions associated with technological characteristics [15,16].

The genetic diversity in yeast from several fermented food products has been assessed using various DNA markers, such as microsatellites [17], Inter Simple Sequence Repeat (ISSR) [18], Randomly Amplified Polymorphic DNA (RAPD) [8], Inter-Primer Binding Site (iPBS) [13,19], and Start Codon Targeted (SCoT) [4]. The ISSR markers target the DNA segment between two identical microsatellite repeat regions facing the opposite direction [20]. The system is easy to perform in classical molecular microbiology laboratories and does not require prior sequence information. Despite sharing similarities with the RAPD-PCR technique, ISSR-PCR is much more reproducible and highly polymorphic in yeast thanks to higher primer lengths, higher amplification temperatures, and a non-random amplification mechanism [21].

Many universal ISSR markers have been chiefly utilized to discriminate yeast species and strain typing by simply generating DNA fingerprints alone or in combination [21–25]. On the other hand, very few studies focused on the intra-specific genetic diversity of *S. cerevisiae* strains from different food matrixes using ISSR markers [18,26,27]. There is still a gap in evaluating the intraspecific diversity of *S. cerevisiae* from sourdough and implying the use of ISSR markers in trait-loci associations. This study aimed to investigate the intraspecific genetic diversity of *S. cerevisiae* isolates of Type I sourdough origin, the technological characteristics significant to produce baked goods, and the trait-loci associations.

## 2. Materials and Methods

### 2.1. Yeast Isolates

A total of 96 endogenous *S. cerevisiae* strains previously isolated from 39 Type I sourdough samples and deposited in Aksaray University Food Microbiology Laboratory as glycerol stocks were used within the concept of this study.

The origin of samples belonged to the Central Anatolia region (CAR; *n* = 17), Black Sea region (BSR; *n* = 12), and Aegean region (AER; *n* = 10). The sampling locations of Type I sourdough samples from which the isolates were obtained are given in Figure 1, where the different colors represent different sampling regions. All the strains were activated in Sabaround Dextrose Broth (SDB; Merck, Darmstadt, Germany) for 24–48 h at 25 °C.

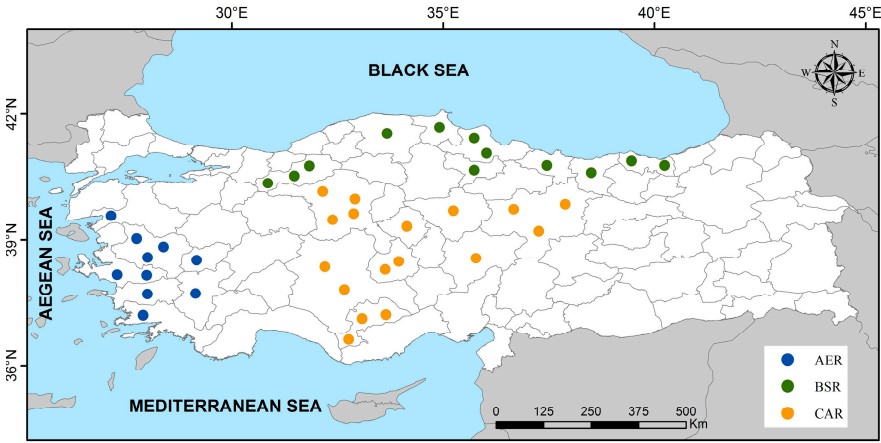

**Figure 1.** The origin of *Saccharomyces cerevisiae* strains. AER: Aegean region, BSR: Black Sea region, CAR: Central Anatolia region.

## 2.2. Molecular Analyses

### 2.2.1. DNA Extraction

According to the manufacturer's directions, DNA was extracted using the DNeasy Blood and Tissue kit (Cat No./ID: 69504, Qiagen GmbH, Hilden, Germany). The DS-11 FX+ spectrophotometer (Denovix Inc., Wilmington, DE, USA) was used to measure the resultant amount of the DNA, and all resultant DNA templates were diluted to 20 ng/μL with sterile ultra-pure water.

### 2.2.2. Molecular Confirmation of the Isolates

Species-specific primer pairs SC1 (5′—AAC GGT GAG AGA TTT CTG TGC—3′) and SC2 (5′—AGC TGG CAG TAT TCC CAC AG—3′) designed by Josepa et al. [28] were employed for the molecular confirmation of *S. cerevisiae*. As a positive control, *S. cerevisiae* 46-Y47 with an NCBI accession number MK358160 was used. The PCR reactions were performed with a 50 μL reaction mixture as previously given in detail in [19]. The PCR products were visualized using a gel imaging system (G:BOX F3, Syngene, Cambridge, UK) under UV after gel electrophoresis on 1.2% (*w/v*) agarose gel and staining with ethidium bromide.

### 2.2.3. ISSR Analyses

Eight ISSR primers, previously reported to have high discriminatory power in yeast, were used to assess intraspecific genetic variation [23–25,27]. Detailed information regarding the primers was presented in Table 1. PCR reactions were carried out in 25 μL reaction mixtures containing 200 μM dNTPs, 0.24 μM primer, 2 mM MgCl2, 1× reaction buffer, and one-unit Taq DNA polymerase (Thermo Fischer Scientific, Waltham, MA, USA). Amplification reactions were performed using a T100 thermocycler as follows: 95 °C for 4 min, followed by 35 cycles of denaturation at 94 °C for 30 s, 30 s of annealing at 45.6–55.0 °C depending on the G/C content of the ISSR primer, and 2 min extension at 72 °C with a final extension step at 72 °C for 10 min. Amplified DNA fragments were subjected to gel electrophoresis on 1.4% (*w/v*) agarose gel, stained with ethidium bromide, visualized using a UV transilluminator using the G: BOX F3 system.

**Table 1.** The list of ISSR markers used and their information.

| ISSR ID | Primer Sequence (5′–3′) | Tm (°C) | TB | PB | PIC | RP |
|---------|-------------------------|---------|-----|-----|------|------|
| ISSR1 | ARRTYCAGCAGCAGCAG | 50.0 | 29 | 23 | 0.66 | 2.72 |
| ISSR2 | GTGGTGGTGGTGGTG | 55.0 | 11 | 5 | 0.76 | 0.27 |
| ISSR3 | AGAGAGAGAGAGAAGAGAGT | 50.4 | 9 | 6 | 0.58 | 0.73 |
| ISSR4 | HVHTGTGTGTGTGTG | 45.6 | 30 | 25 | 0.79 | 3.25 |
| ISSR5 | AGAGAGAGAGAGAGAGVC | 50.8 | 25 | 23 | 0.44 | 1.46 |
| ISSR6 | ACACACACACACAGACYG | 54.0 | 10 | 7 | 0.59 | 0.86 |
| ISSR7 | AGAGAGAGAGAGAGAGYG | 51.4 | 23 | 20 | 0.74 | 2.16 |
| ISSR8 | GACAGACAGACAGACA | 52.0 | 17 | 14 | 0.58 | 1.31 |

Tm: Annealing Temperature, TB: Total Band, PB: Polymorphic Band, PIC: Polymorphism Information Content, RP: Resolving Power.

### 2.2.4. Statistical Evaluation of Molecular Analyses

The reproducibility of the amplification achieved using each of the eight primers was guaranteed. To create a binary data matrix, strong, clear, and unambiguous PCR bands were carefully rated as present (1) or missing (0) at their locations. A 100 bp Low Ladder (P1473; Sigma–Aldrich, St. Louis, MO, USA) was used as a molecular weight identifier to gauge the size of the PCR products. Each ISSR marker's performance was calculated using the resolving power (RP) and the polymorphic information content (PIC)

in EXCEL (Microsoft, Redmond, WA, USA), as proposed by Prevost and Wilkinson [29] and Roldàn-Ruiz et al. [30].

The binary data matrix was converted into a genetic similarity matrix using Jaccard's similarity coefficient, and Unweighted Pair Group Method Using Arithmetic Average (UPGMA) was constructed with the 'vegan' package of 'R Studio' software [31,32].

The isolates were divided according to sampling region (*n* = 3), CAR, BSR, and AER. The binary data were analyzed using GenAlex 6.5 [33] and PopGene [34] to estimate the observed and effective number of alleles, Nei's gene diversity, and Shannon's information index. Analysis of molecular variance (AMOVA) was conducted to reveal the genetic variation between and within the populations with 999 permutations [35].

STRUCTURE v.2.3.4 was used to analyze the binary data matrix without prior population origin information to test the best $K$ ($\Delta K$) value, which provides the most significant sub-groups within all *S. cerevisiae* populations [36]. Ten iterations were chosen for each of the $K$ values to be tested, which ranged from 2 to 10. Both the length of burning and the number of Markov Chain Monte Carlo (MCMC) repeats after burning were set to 100.000 [37]. The results were uploaded to the STRUCTURE HARVESTER web-based analysis tool to estimate the $\Delta K$ using the Evanno method [38].

### 2.3. Technological Characterization

The following technological characteristics were evaluated in 29 endogenous *S. cerevisiae* strains that were genetically discriminated by the molecular assessment. All technological assessments were performed in duplicate to assure the statistical data.

### 2.3.1. Growth at Different pH Values, NaCl Concentrations and Temperatures

The SDB inoculated with 3 CFU/mL of yeast were first incubated at 10, 25, and 37 °C for 48 h. The SDB adjusted to different pH values (2.5, 3.0, 3.5) and containing different NaCl concentrations (2, 4, and 6%) were inoculated with 3 CFU/mL to be incubated for 48 h at 25 °C. After incubation, the microbial growth was measured using a Allsheng AMR-100T spectrophotometer (Hangzhou Allsheng, Instruments Co. Ltd., Hangzhou, China). Aliquots of non-fortified SAB broth inoculated with 3 CFU/mL of yeasts and incubated at 25 °C were used as positive controls. The data were arranged as Growth Index (GI) using the following equation [39]:

$$GI = (Abs_s/Abs_c) \times 100 \tag{1}$$

where Abss represents the absorbance of the yeast isolates at different pH values, NaCl concentrations, and the incubation temperatures and the control group, respectively. Absc, on the other hand, stands for the control group. GI values were regarded as follows [39,40]:

$$GI < 25\% \text{ strong inhibition}$$

$$25\% < GI < 75\% \text{ moderate inhibition}$$

$$GI > 75\% \text{ high growth}$$

### 2.3.2. Resistance to Lactic Acid and Acetic Acid

The SDB fortified with lactic acid (0.6 and 1.2%), and acetic acid (0.15 and 0.30%) were inoculated with 3 CFU/mL of yeast to be incubated at 25 °C for 48 h. The microbial growth was determined through absorbance measurement, and the results were given as GI (see Section 2.3.1). The non-fortified SDB inoculated with 3 CFU/mL of yeast was used as a control [40].

### 2.3.3. Fermentation Rate

The fermentation rate assay proposed by Pérez-Coello et al. [41] was used with minor modifications. The activated yeast cultures were inoculated ($10^6$ CFU/mL) to 200 mL of

SDB containing 20% glucose. Fermentation flasks were plugged with glass fermentation traps containing sterile distilled water to allow only $CO_2$ to exit. The fermentation was carried out at 25 °C till the end, and the weight loss was measured daily. The weight loss during the fermentation and the weight loss between the 24th and 72nd hours of fermentation were calculated separately. The results were given as FCx ($gCO_2$/day) and FCy ($gCO_2$/L·h).

FCx < 3.75 and FCy < 0.80; weak fermentation rate

3.75 < FCx < 4.0 and 0.80 < FCy < 1.00; moderate fermentation rate

FCx > 4.00 and FCy > 1.00; high fermentation rate

2.3.4. Leavening Activity

The activated yeast cells were washed and suspended in sterile tap water. For each yeast strain, 100 g Type 0 wheat flour, 60 mL sterile tap water, 0.14 g NaCl and yeast cells ($10^6$ CFU/mL) were mixed in sterile graduated containers. The samples were incubated at 25 °C for 5 h. The volume increase was measured hourly. The data were modelled as Volume Index (VI) values using the following equation:

$$VI = ((Vf - Vi)/Vi) \times 100 \qquad (2)$$

where the Vf implies the volume of the dough obtained after 5 h, and Vi stands for the initial volume. VI values were regarded as follows [40]:

VI = 0; no leavening activity

VI < 50; weak leavening activity

VI > 50; strong leavening activity

2.3.5. Statistical Analyses

A multivariate approach evaluated statistically prominent *S. cerevisiae* strains. For this purpose, Principal Component Analyses (PCA) was employed using JMP Pro 16.0 software (trial version) (SAS Institute Inc, Cary, NC, USA). The quantitative values were converted into qualitative codes (0, 1, 2) as implied in Table 2. The convenience of the data for PCA was tested according to Bartlett's test [42]. Additionally, the discriminative effects of the studied circumstances on the strains were evaluated by utilizing a heatmap analysis according to Ward's method in the JMP Pro 16.0 software.

**Table 2.** Qualitative codes for multivariate analysis (PCA).

| Quantitative Analyses [a] | | | | Codes |
|---|---|---|---|---|
| [b] GI < 25 | FCx < 3.75 | FCy < 0.80 | VI = 0 | 0 |
| 25 < GI < 75 | 3.75 < FCx < 4.0 | 0.80 < FCy < 1.00 | VI < 50 | 1 |
| GI > 75 | FCx > 4.0 | FCy > 1.0 | VI > 50 | 2 |

[a] Growth at different pH values, NaCl concentrations and temperatures, resistance to lactic acid and acetic acid, fermentation capacity, and leavening activity. [b] GI: Growth Index; VI: Volume Index; FCx: Fermentation Capacity ($gCO_2$/day); FCy: Fermentation Capacity ($gCO_2$/L·h).

2.4. *Trait-Loci Associations*

Trait-loci association analyses were performed using the TASSEL 5 software, where the general linear model (GLM) incorporating the PCA and the STRUCTURE (q) results were used [43,44]. Association between the traits and loci was considered significant when $p < 0.05$.

## 3. Results

### 3.1. Intraspecific Genetic Variation

The eight ISSR primers amplified 154 reproducible and scorable bands, of which 79.87% were polymorphic. The amplified DNA fragments for each ISSR primer ranged from nine (ISSR-3) to 30 (ISSR-4), with a total ratio of 19.25 per primer. Degenerate primers produced more scorable total bands in general. The PIC and RP values indicating the effectiveness of the primers are given in Table 1. The highest PIC value was obtained from the ISSR-2 primer (0.76), whereas the lowest PIC value was obtained from the ISSR-5 primer (0.44). On the other hand, ISSR-4 and ISSR-2 primers yielded the highest and lowest RP values of 3.25 and 0.27, respectively.

The genetic diversity indices were also determined by expressing the observed number of alleles (1.81 ± 0.09), the effective number of alleles (1.42 ± 0.04), Nei's gene diversity (0.24 ± 0.01), and Shannon's information index (0.36 ± 0.02). These values were also determined within the populations. Accordingly, the individuals of CAR and BSR populations were the wealthiest in allelic richness. They were genetically more diverse according to the Shannon's index and Nei's gene diversity indices, whereas these values were the lowest for the AER population. The percentage of the polymorphic loci values also supported these values (Table 3).

**Table 3.** Genetic variation of different populations obtained by ISSR markers.

| Population | Na ** | Ne | *I* | *h* | PPL (%) |
|---|---|---|---|---|---|
| CAR * (*n* = 32) | 1.60 ± 0.05 | 1.37 ± 0.03 | 0.33 ± 0.02 | 0.22 ± 0.01 | 69.48 |
| BSR (*n* = 32) | 1.56 ± 0.06 | 1.40 ± 0.03 | 0.33 ± 0.02 | 0.22 ± 0.01 | 68.18 |
| AER (*n* = 32) | 1.32 ± 0.06 | 1.31 ± 0.03 | 0.28 ± 0.02 | 0.19 ± 0.01 | 53.25 |
| Average (*n* = 96) | 1.50 ± 0.03 | 1.36 ± 0.02 | 0.31 ± 0.01 | 0.21 ± 0.01 | 63.64 ± 5.21 |

* CAR: Central Anatolia Region, BSR: Black Sea Region, AER: Aegean Region. ** Na: The number of alleles, Ne: The effective number of alleles, *I*: Shannon's information index, *h*: Nei's gene diversity, PPL (%): Percentage of Polymorphic Loci.

The AMOVA revealed significant ($p < 0.001$) genetic differences between and within the populations, as given in Table 4. Of the total genetic variation, most of the genetic variation was observed within the populations (85%), which is also supported by the low $F_{ST}$ value (0.155). Additionally, the values for gene flow ($N_m$) and genetic variation ($G_{ST}$) were found to be 0.13 and 3.24, respectively.

**Table 4.** AMOVA results of *Saccharomyces cerevisiae* populations.

| Source | *d-f* | SS | MS | Est. var. | % | $F_{ST}$ | *p* |
|---|---|---|---|---|---|---|---|
| AP * | 2 | 230.375 | 115.188 | 3.075 | 15 | | |
| AIWP | 93 | 1562.500 | 16.801 | 16.801 | 85 | 0.155 | 0.001 |
| Total | 95 | 1792.875 | | 19.876 | 100 | | |

* AP: Among Populations, AIWP: Among Individuals Within Populations, *d-f*: degrees of freedom, SS: Sums of Squares, MS: Mean Square, Est.var.: Estimated Variation, $F_{ST}$: Fixation Index, *p*: Significance Level.

The isolates were grouped as two main clusters on the UPGMA dendrogram (Figure 2). Cluster I grouped 77 strains, while Cluster II grouped the rest. There was no clear genetic differentiation among the populations according to the geographic regions on the UPGMA dendrogram, which is also indicated on the PCoA dendrogram (Figure 3). However, closer geographical locations within the same populations most likely tended to group more closely, such as Y3, Y4 and Y21, Y22 as well as Y87, Y88 and Y65, Y66. The data had the highest probability conducted by the Bayesian clustering model on STRUCTURE when the individuals were split into two populations ($\Delta K = 2$), implying two statistically significant clusters as shown on the UPGMA.

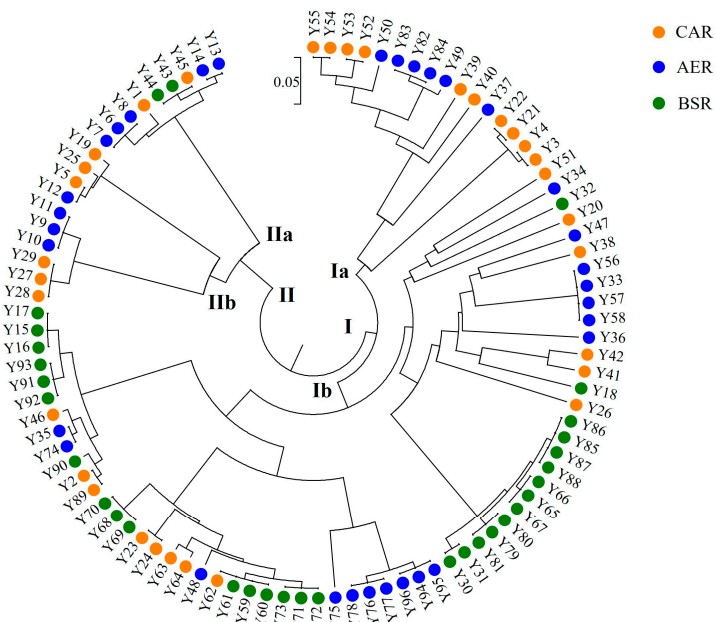

**Figure 2.** The UPGMA dendrogram of 96 *Saccharomyces cerevisiae* isolates based on Jaccard's coefficient. The bold letters (I, II, Ia, Ib, IIa, and IIb) indicate the sub-groups.

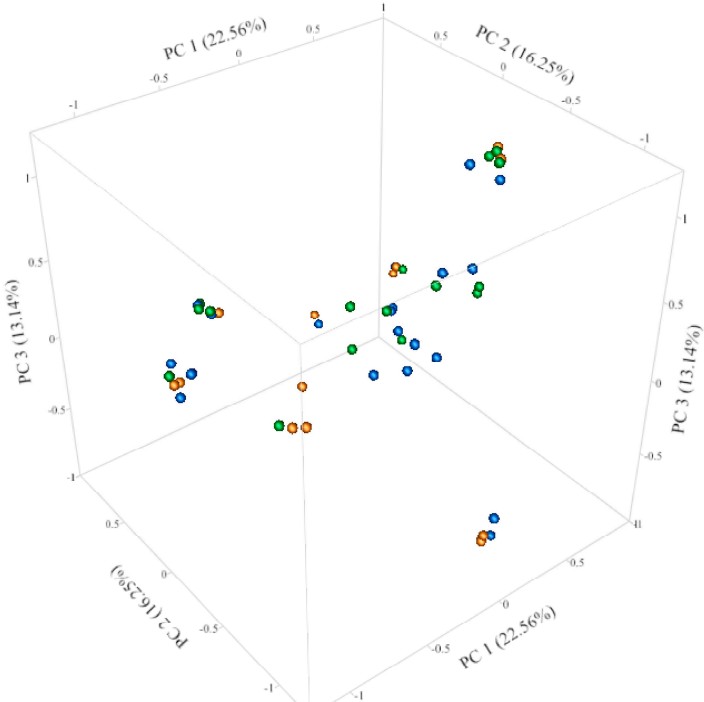

**Figure 3.** Three dimensional PCoA distribution of the isolates. Different colors represent different sampling regions. Central Anatolia region (CAR): Orange circle. Black Sea region (BSR): Green circle. Aegean region (AER): Blue circle.

### 3.2. Technological Characterization

As seen in Figure 2, most sub-clusters have more than one isolate. We chose one isolate for each sub-cluster to select the genetically different isolates. Accordingly, genetically diverse 29 *S. cerevisiae* isolates representing each sub-cluster on the UPGMA dendrogram (Figure 2) were chosen to be technologically analyzed. The new UPGMA dendrogram of the selected isolates is also given in Figure 4, with a discriminatory power of 0.98.

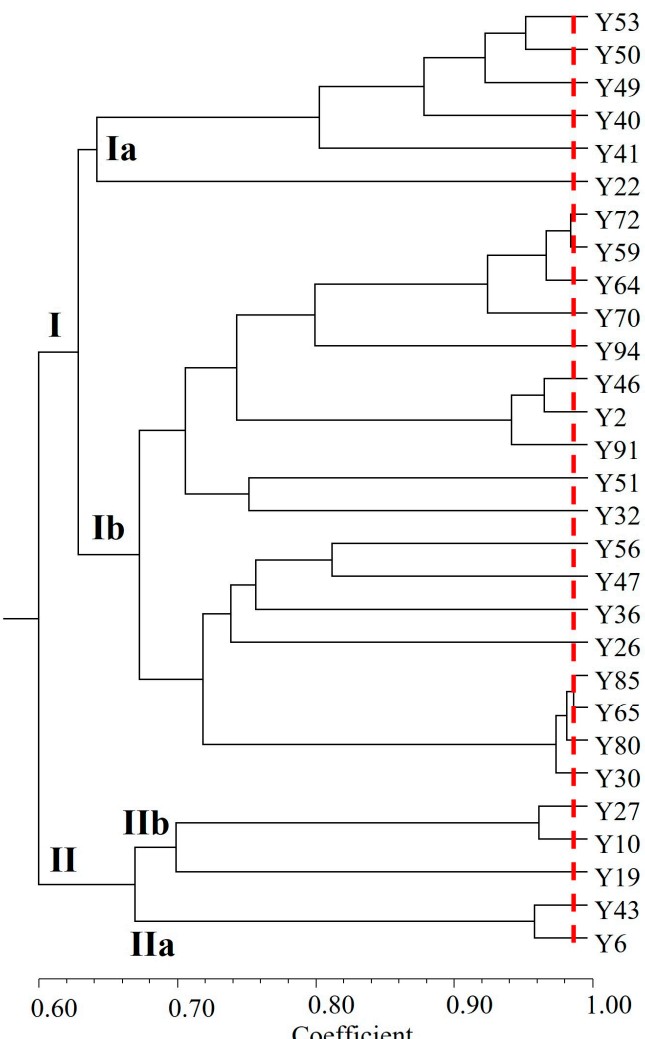

**Figure 4.** The UPGMA dendrogram of 29 genetically diverse *Saccharomyces cerevisiae* strains. The bold letters (I, II, Ia, Ib, IIa, and IIb) indicate the sub-groups.

The overall technological characterization results are presented in Table 5. Twenty-two (75.86%) strains were partially inhibited (25 < GI < 75) when incubated at 10 °C, whereas sixteen (55.17%) strains exhibited a growth pattern as control (GI > 75) at 37 °C. Regarding the salt resistance, its effect mainly depended on the NaCl concentration. All the strains grew as control when the concentration was 2%. On the other hand, increasing concentration significantly affected yeast growth. The growth of 10 strains (34.48%) was wholly inhibited (GI < 25) at 6% salt concentration.

Among all of these, three strains (Y43, Y70, and Y91) were partially inhibited (25 < GI < 75) at pH 3.5, which is an expected pH of the sourdough environment. Lower pH also inhibited the growth of many strains. Most of the strains (79.31%) were utterly inhibited at pH 2.5, whereas six strains exhibited moderate growth (25 < GI < 75), which can later be significant for the probiotic characteristic studies. Low lactic acid (0.60%) and acetic acid (0.15%) concentrations did not notably affect most strains' growth. However, increasing concentrations gave rise to moderate inhibition (25 < GI < 75). At increasing acid concentrations, none of the strains were inhibited entirely (GI > 25). Increasing acetic acid concentrations were more likely to inhibit more strains.

The results regarding the fermentation based on $CO_2$ loss per day (g$CO_2$/day and g$CO_2$/L·h) are summarized in Table 5. The total weight loss based on $CO_2$ for each strain ranged between $17.41 \pm 0.11$ g$CO_2$ (*S. cerevisiae* Y30) and $19.79 \pm 0.08$ g$CO_2$ (*S. cerevisiae* Y85) with a median value of $18.52 \pm 0.87$ g$CO_2$. The fermentation rate was based on the

total $CO_2$ loss per day ($gCO_2$/day) and $CO_2$ loss, where the fermentation rate was the highest between the 24th and 72nd hours of fermentation divided by fermentation volume ($gCO_2$/L·h). Accordingly, 62.06% of the strains (*n* = 18) displayed a high fermentation rate (FCx > 4.0) based on $gCO_2$/day. On the other hand, 51.72% of the strains (*n* = 15) with high or moderate (FCx > 3.75) $gCO_2$/day value exhibited a high fermentation rate (FCy > 1.0) based on $gCO_2$/L·h. The leavening ability of the selected *S. cerevisiae* strains in a certain time ranged from 41.09 ± 2.34 (*S. cerevisiae* Y6) to 89.14 ± 3.52 (*S. cerevisiae* Y85) as percentages. A total of 23 strains displayed a strong leavening activity (VI > 50), whereas the remaining strains resulted in weak leavening activity (VI < 50).

**Table 5.** Technological characteristics of isolates.

| No | [a] Temperature (°C) | | NaCl | | | pH | | | LAR (%) | | AAR (%) | | FCx | FCy | Lev |
|----|------|------|------|------|------|------|------|------|------|------|------|------|------|------|------|
| | 10 | 37 | 4 | 6 | 2.5 | 3.0 | 3.5 | 0.60 | 1.20 | 0.15 | 0.30 | | | |
| Y2 | 25–75 | >75 | 25–75 | 25–75 | <25 | 25–75 | >75 | >75 | >75 | >75 | >75 | >4.0 | >1.0 | >50 |
| Y6 | 25–75 | 25–75 | 25–75 | 25–75 | <25 | 25–75 | >75 | >75 | 25–75 | >75 | 25–75 | 3.5–3.75 | <0.80 | <50 |
| Y10 | 25–75 | 25–75 | 25–75 | <25 | <25 | 25–75 | >75 | >75 | >75 | >75 | >75 | 3.5–3.75 | 0.80–1.00 | <50 |
| Y19 | 25–75 | 25–75 | >75 | 25–75 | <25 | 25–75 | >75 | >75 | >75 | >75 | >75 | 3.75–4.0 | 0.80–1.00 | >50 |
| Y22 | 25–75 | 25–75 | 25–75 | 25–75 | <25 | 25–75 | >75 | >75 | 25–75 | >75 | >75 | >4.0 | >1.0 | >50 |
| Y26 | 25–75 | >75 | 25–75 | <25 | <25 | 25–75 | >75 | >75 | >75 | >75 | 25–75 | 3.75–4.0 | >1.0 | >50 |
| Y27 | 25–75 | 25–75 | >75 | 25–75 | <25 | 25–75 | >75 | >75 | >75 | >75 | >75 | >4.0 | >1.0 | >50 |
| Y30 | 25–75 | >75 | >75 | 25–75 | <25 | 25–75 | >75 | >75 | 25–75 | 25–75 | 25–75 | >4.0 | >1.0 | >50 |
| Y32 | 25–75 | 25–75 | 25–75 | 25–75 | <25 | 25–75 | >75 | >75 | >75 | >75 | >75 | 3.5–3.75 | <0.80 | <50 |
| Y36 | >75 | 25–75 | 25–75 | 25–75 | <25 | 25–75 | >75 | >75 | >75 | >75 | >75 | >4.0 | 0.80–1.00 | >50 |
| Y40 | 25–75 | >75 | 25–75 | <25 | <25 | 25–75 | >75 | >75 | >75 | 25–75 | 25–75 | 3.75–4.0 | >1.0 | >50 |
| Y41 | 25–75 | 25–75 | 25–75 | <25 | <25 | 25–75 | >75 | >75 | >75 | >75 | 25–75 | >4.0 | 0.80–1.00 | >50 |
| Y43 | 25–75 | >75 | 25–75 | <25 | <25 | 25–75 | 25–75 | 25–75 | 25–75 | >75 | 25–75 | 3.75–4.0 | >1.0 | >50 |
| Y46 | 25–75 | >75 | 25–75 | <25 | <25 | 25–75 | >75 | >75 | >75 | >75 | >75 | >4.0 | >1.0 | >50 |
| Y47 | 25–75 | 25–75 | >75 | 25–75 | <25 | 25–75 | >75 | >75 | >75 | >75 | >75 | 3.5–3.75 | <0.80 | <50 |
| Y49 | 25–75 | >75 | 25–75 | <25 | <25 | 25–75 | >75 | >75 | >75 | >75 | 25–75 | 3.5–3.75 | 0.80–1.00 | <50 |
| Y50 | 25–75 | >75 | >75 | 25–75 | 25–75 | >75 | >75 | >75 | >75 | >75 | >75 | >4.0 | >1.0 | >50 |
| Y51 | >75 | 25–75 | >75 | 25–75 | <25 | 25–75 | >75 | >75 | >75 | >75 | >75 | >4.0 | 0.80–1.00 | >50 |
| Y53 | >75 | >75 | >75 | 25–75 | 25–75 | >75 | >75 | >75 | >75 | >75 | >75 | 3.5–3.75 | 0.80–1.00 | <50 |
| Y56 | 25–75 | >75 | 25–75 | 25–75 | <25 | 25–75 | >75 | >75 | >75 | 25–75 | 25–75 | >4.0 | >1.0 | >50 |
| Y59 | >75 | >75 | 25–75 | 25–75 | <25 | 25–75 | >75 | >75 | >75 | >75 | 25–75 | >4.0 | >1.0 | >50 |
| Y64 | 25–75 | >75 | 25–75 | 25–75 | 25–75 | >75 | >75 | >75 | >75 | >75 | >75 | >4.0 | >1.0 | >50 |
| Y65 | >75 | >75 | >75 | 25–75 | 25–75 | 25–75 | >75 | >75 | >75 | >75 | >75 | 3.5–3.75 | 0.80–1.00 | >50 |
| Y70 | 25–75 | 25–75 | 25–75 | <25 | <25 | 25–75 | 25–75 | 25–75 | 25–75 | >75 | >75 | >4.0 | 0.80–1.00 | >50 |
| Y72 | >75 | >75 | >75 | 25–75 | 25–75 | >75 | >75 | >75 | 25–75 | >75 | >75 | >4.0 | >1.0 | >50 |
| Y80 | 25–75 | 25–75 | 25–75 | <25 | <25 | 25–75 | >75 | >75 | >75 | >75 | >75 | >4.0 | 0.80–1.00 | >50 |
| Y85 | >75 | >75 | >75 | 25–75 | 25–75 | >75 | >75 | >75 | >75 | >75 | >75 | >4.0 | >1.0 | >50 |
| Y91 | 25–75 | >75 | 25–75 | <25 | <25 | 25–75 | 25–75 | >75 | 25–75 | 25–75 | 25–75 | >4.0 | >1.0 | >50 |
| Y94 | 25–75 | 25–75 | 25–75 | 25–75 | <25 | 25–75 | >75 | >75 | >75 | >75 | >75 | >4.0 | 0.80–1.00 | >50 |

[a] Temp: Temperature; LAR: Lactic Acid Resistance; ASR: Acetic Acid Resistance; FCx: Fermentation Capacity as $gCO_2$/day; FCy: Fermentation Capacity as $gCO_2$/L·h; Lev: Leavening Activity.

The heatmap analysis result indicates that the fermentation rate, leavening activity, and growth ability at 37 °C seem parallel and separated from the other traits. On the other hand, there was no other apparent relationship between the other technological characteristics. The strains were divided into two main clusters, mainly separated by Temp10, pH 2.5, pH 3.0, and NaCl4 traits. Y50, Y72, and Y85 were entirely equivalent in response to the medium. The identical characteristics in the small cluster were fermentation rates (FCx and FCy) and leavening activity (LeV), while the main discriminants in the big group were growth at 37 °C and fermentation rates (FCx and FCy) (Figure 5).

As a final step for technological characterization, we used a multivariate approach to assess the technologically prominent *S. cerevisiae* strains. The factor 1 and factor 2 loadings were found to be statistically significant (*p* < 0.0001) and suitable for the PCA according to the Bartlett test criteria [42]. The PCA exhibited 49.2% variability, emphasizing a homogenous distribution with significant differences (Figure 6). The variables that the factors used most effectively were identified by examining the factor loads. When the eigenvalues for each factor were calculated, factors 1 (3.737) and 2 (2.896) had the highest values. Factor 1 consisted of growth variables at different acetic and lactic acid concentrations, pH 3.5, and 6% NaCl. On the other hand, Factor 2 included leavening activity, fermentation capacity, and growth ability at 10 and 37 °C, pH 3.0, 3.5, and 4% of NaCl. Seven strains (Y2, Y27, Y50, Y59, Y64, Y72, and Y85) were technologically superior to

others by having high fermentation capacity, leavening activity, and growth index at 37 °C. Among them, Y50, Y64, Y72, and Y85 formed another sub-group, whose factor loadings are the highest due to higher growth abilities at lower pH values, 4% NaCl, and 10 °C.

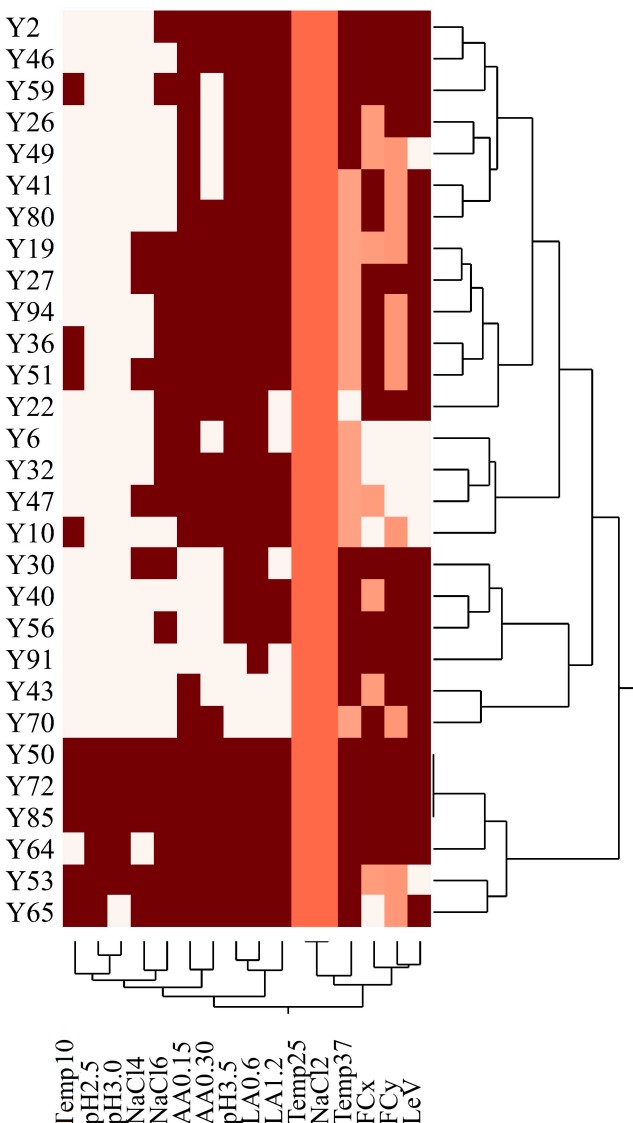

**Figure 5.** The heatmap analysis scheme of the technological traits and the isolates (*n* = 29). The color scale from dark red to white indicates high to low values in technological traits.

### 3.3. Trait-Loci Associations

Although there was no apparent relationship between the genotypes and technological characteristics according to the UPGMA dendrogram, we used TASSEL 5 statistical software to find the statistically significant trait-loci associations ($p < 0.05$). For this purpose, the GLM was used, with the results given in Figure 7.

According to GLM, the highest correlation ($r = 0.60$) was recorded for the fermentation capacity (FCy; $gCO_2/L \cdot h$) trait on the 8th and 113th loci, amplified by ISSR-1 and ISSR-6 primers, respectively. The PCR fragments obtained using ISSR-6 is given in Figure 8. The strains were randomly selected and amplified using the ISSR-6 primer. The polymorphic band specified with a red arrow marker around 800 bp was most associated with the fermentation capacity according to GLM (Y2, Y27, Y43, Y49, Y61, Y72, Y80, Y91). To ensure the reproducibility of the ISSR-6 primer, different DNA concentrations (20, 30, 40, and 50 ng) were used for the amplification, as proposed by Aydın et al. [4]. As well as this, the

fragments were robust at different DNA concentrations. In addition, none of the traits were found to be polygenic according to ISSR-PCR.

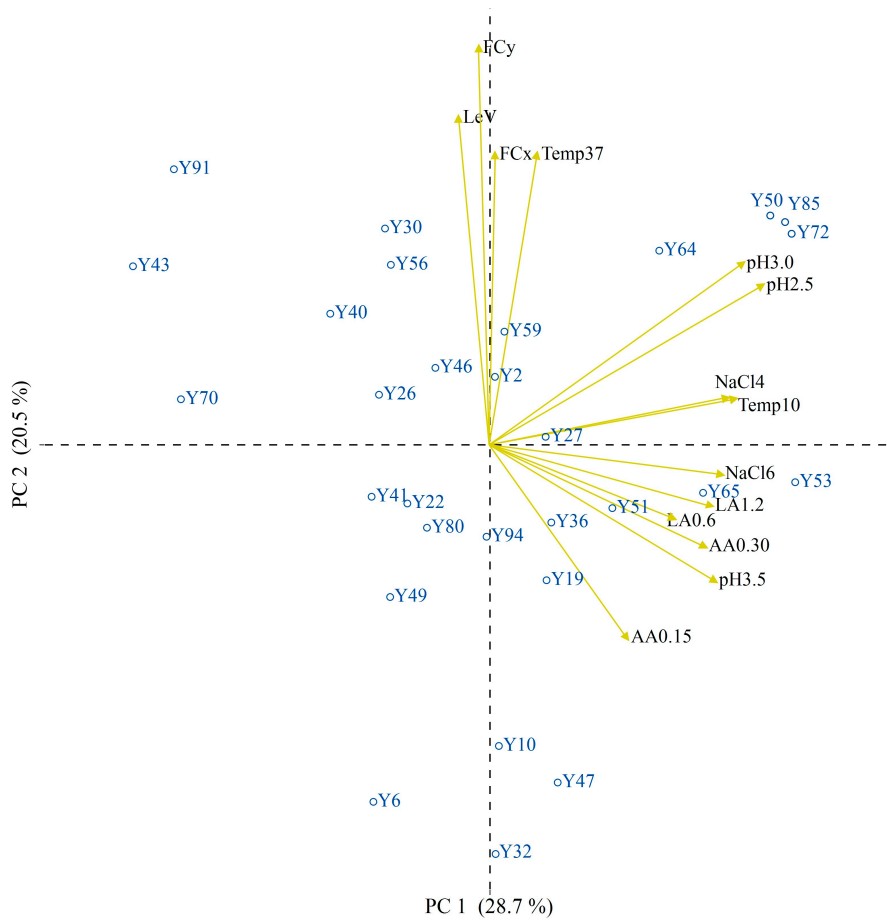

**Figure 6.** Principal Component Analysis of 29 genetically diverse *Saccharomyces cerevisiae* strains for the selection of the most relevant technological isolates. AA: acetic acid; LA: lactic acid; NaCl, Temp: Temperature; LeV: leavening activity; FCx: fermentation capacity as $gCO_2/day$; FCy: fermentation capacity as $gCO_2/L \cdot h$.

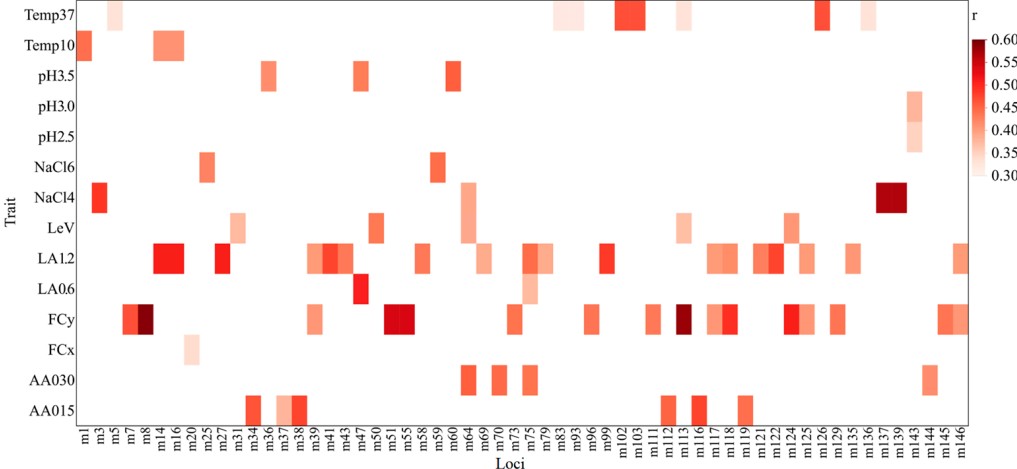

**Figure 7.** Trait-loci association analyses according to the general linear model. AA: acetic acid, LA: lactic acid, Temp: Temperature, LeV: leavening activity, FCx: fermentation capacity as $gCO_2/day$, FCy: fermentation capacity as $gCO_2/L \cdot h$.

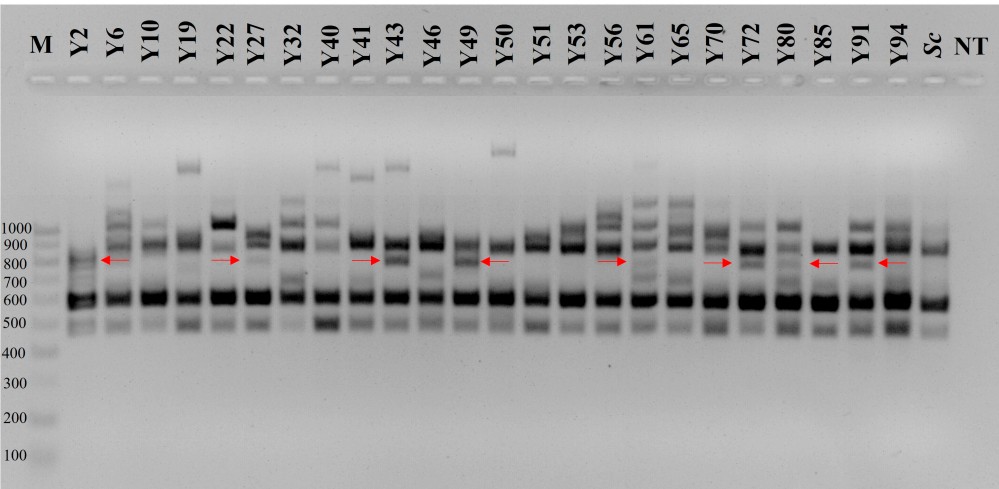

**Figure 8.** PCR banding patterns of different strains obtained by ISSR-6 primer. The red arrow indicates the polymorphism associated with the fermentation rate according to general linear model. M: PCR 100 bp Low Ladder (P1473; Sigma-Aldrich, St. Louis, MO, USA), Sc: *Saccharomyces cerevisiae* 46-Y47 with an NCBI accession number MK358160, NT: Non-template DNA.

## 4. Discussion

Sourdough yeasts are significant members of the sourdough microbiota along with the LAB. Based on their heterogeneity, endogenous yeast isolated from traditional sourdough may possess various technological characteristics [12]. ISSR markers have successfully been used to detect the polymorphic yeast strains from the sourdough environment [45–47]. Here, we used eight ISSR markers for evaluating the intraspecific genetic diversity of 96 *S. cerevisiae* strains, determined genetically diverse strains' technological characteristics significant for baked goods, and assessed the trait-loci associations. This is the first study where the ISSR markers were used to deeply study the intraspecific genetic diversity of *S. cerevisiae* isolated from sourdough and to investigate the trait-loci associations statistically by TASSEL.

Different parameters, including the percentage of polymorphic loci (PPL), the total and effective number of alleles, Nei's gene diversity, and Shannon's information index are frequently used to assess the intraspecific genetic diversity. The genetic parameters obtained by eight different ISSR markers are comparable to those reported by Liu et al. [18], where the genetic diversity of industrial brewing *S. cerevisiae* strains was evaluated using 15 ISSR primers. On the other hand, these values were slightly higher than those reported by Aydın et al. [19] using iPBS markers in *S. cerevisiae* isolated from Type I sourdough. The results suggest that markers targeting the DNA segment between two tandem repeats in different directions reflect the genetic diversity better than those targeting the retrotransposons. ISSR and iPBS markers should be used on the same isolates within the same study to compare the intraspecific genetic diversity parameters and the Cophenity matrix.

The percentage of polymorphic loci (79.87%) was higher than those reported by Luan et al. [16] but lower than those reported by Liu et al. [18] and Pathania et al. [27], probably due to the lower ratio of degenerate primers used in this study. The PIC value reflects the ability of a primer to evaluate genetic variation and is classified as follows: (i) PIC > 0.50, high polymorphism; (ii) 0.50 > PIC > 0.25, moderate polymorphism; and (iii) PIC < 0.25, low polymorphism [48]. The PIC values obtained from each ISSR primer differed between 0.44 (ISSR-5) and 0.79 (ISSR-4), implying a high intraspecific genetic polymorphism, which can be attributed to the strains' widespread geographic origin.

The CAR and BSR populations were genetically more diverse according to their allelic richness, Nei's gene diversity, and Shannon's information index than those in the AER population. According to Aydın et al. [4], the lowest polymorphism was detected in CAR populations, whereas the genetic variation in AER individuals was more diverse

using SCoT markers. Similarly, iPBS markers gave different genetic variation indexes within the populations. These dominant markers should be used together to enhance the resolution of genetic variation parameters. AMOVA results suggested that most of the genetic variation (85%) were detected within individuals among the populations. The higher genetic difference within these populations offers a higher subdivision level and hierarchy [49]. The results are in accordance with those reported by other researchers using iPBS and SSR markers to assess intraspecific genetic variation in *S. cerevisiae* isolated from sourdough and wine [19,50,51].

The UPGMA divided 96 endogenous *S. cerevisiae* strains into two main clusters, supported by the STRUCTURE output ($\Delta K = 2$). According to the STRUCTURE output, nine strains (Y32, Y47, Y51, Y76, Y77, Y78, Y94, Y95, Y96) were equally close to both clusters due to having many more shared alleles. This could arise from the shared alleles between strains from different geographic origins, which occur less frequently, and is supported by low gene flow value [52–54]. The genetic differentiation did not occur according to geographic region according to the UPGMA dendrogram and PCoA distribution. A possible reason could be the transportation of sourdough samples between different locations through online shopping. Especially since the beginning of the COVID pandemic, many people have made sourdough and sold it through social media in Türkiye.

Selecting technologically prominent yeast strains is a difficult task due to the excess of data, which requires clustering and statistical techniques. Obtaining genetically diverse genotypes permits the selection according to genetic structure, and this technique can be applicable when prior information regarding the strains is unavailable [14]. Numerous researchers used inter-delta, RAPD, and mtDNA-RFLP markers to lower the number of isolates by simply obtaining genetically diverse genotypes before technological characterization [8,55–58]. The ISSR markers separated different biotypes with a discriminatory power of 0.98, higher than those reported by Palla et al. [17] and Palla et al. [58] using microsatellite markers for strain typing of sourdough yeasts.

Pulvirenti et al. [59] suggested a procedure for a starter culture selection, as follows: (i) The isolation and identification of the dominating strains; (ii) evaluating the technologically relevant strains; and (iii) the selection of the strains to produce desired fermented food product. The selection of candidate yeast starters for baked goods is based on various technological factors, including their ability to leaven dough, produce $CO_2$, and resist difficult growth conditions [60]. In our study, the selection of the technologically relevant isolates depended on the harsh growth conditions, fermentation rate, and strains' ability to leaven the dough. Although some factors are not strictly related to producing baked goods, such as low pH and high salt concentrations, the strains were expected to overcome these harsh conditions, as sourdough is a stressful environment brought about by the competing LAB and yeast. The strains were considered technologically relevant when they grew as control (GI > 75). Resistance to lactic acid and acetic acid is a key factor since some yeast strains may be affected by the lactic and acetic acid produced by the competing LAB or yeast within the microbiota of sourdough. The results obtained are in accordance with *S. cerevisiae* strains isolated from Altamura sourdough with minor differences [40].

The most significant characteristics required for yeast starters are leavening ability and high fermentation rate. Economically, bread makers profit from *S. cerevisiae* strains' high leavening capacity. *S. cerevisiae* strains ferment the sugar into $CO_2$, ethanol, and glycerol, thus leavening the dough [61]. The heatmap analysis indicated that the leavening activity and fermentation rate results are primarily parallel. Still, there are some strains (Y19, Y51, Y65, Y70, Y80, Y94) with high leavening activity (VI > 50) and moderate fermentation rate (1.00 > FCy > 0.80). The ability of *S. cerevisiae* strains to leaven is influenced by additional variables besides their capacity for glucose fermentation. *Saccharomyces cerevisiae* strains primarily ferment glucose, whereas maltose and fructose are also fermented at later stages of the fermentation [62]. The leavening ability is also affected by sucrase activity and osmotolerance; however, the underlying mechanism requires further investigation [63]. Similar findings were also reported by Yang et al. [8].

The multivariate approaches provided a strain grouping according to technological traits' input data and pointed out the variables responsible for the differentiation. Among 29 *S. cerevisiae* strains, five were found to be the technologically most relevant isolates according to PCA distribution (Figure 6). These strains can be used as yeast starter candidates to produce baked goods. Sourdough is also used to make shalgam and cereal-based traditional fermented drinks, such as boza [64,65]. The technologically most relevant strains' probiotic characteristics should be further investigated for producing cereal-based probiotic food products.

The associations between genotypes and technological traits, according to the UPGMA dendrogram, are insufficient, since a single polymorphic band could be significant for a specific trait. The complex statistical methods examining the possible associations between the loci and traits are gaining more interest. We used the GLM approach to seek statistically significant associations ($p < 0.05$). The results indicate that the ISSR-6 primer may be valuable for detecting strains with high fermentation rates (FCy; $gCO_2/L \cdot h$). As mentioned above, the strains with high leavening activity and fermentation rate are of particular interest for producing baked goods. The ISSR markers are length polymorphic markers, where each locus is visualized on the agarose gel. Producing SCAR markers using the trait-associated locus (Figure 8) detected in this study is possible for strain identification with a high fermentation rate. The utility of ISSR markers to detect significant trait-loci associations has been reported by several authors in plants [66,67]. We hereby introduce significant trait–loci associations ($p < 0.05$) in *S. cerevisiae* using the ISSR markers for the first time. ISSR markers may also be used for different yeast species for further trait-loci associations.

## 5. Conclusions

The genetic variation of *S. cerevisiae* strains from traditional sourdough was investigated by eight ISSR primers. They helped reveal the genetic variation between and within populations and genetically diverse *S. cerevisiae* isolates before technological characterization. The multivariate approach assessed the technologically relevant strains as candidate starters to produce baked goods. According to the trait–loci associations, a significant association ($p < 0.05$) was found between a polymorphic locus and the fermentation rate. ISSR markers should also be used in different yeast species to define trait–loci associations further. Our future studies will cover producing SCAR markers targeting the specific region related to fermentation rate and further definition of trait-associated loci by using different DNA markers, such as iPBS and SCoT.

**Author Contributions:** Conceptualization: F.A., G.Ö. and İ.Ç.; Methodology: F.A., H.İ.K., İ.Ç., G.Ö. and E.G.; Investigation: F.A., H.İ.K., T.U.G. and Y.A.; Validation: E.G., G.Ö. and İ.Ç.; Writing—Original draft preparation: F.A.; Writing—review & editing: F.A., G.Ö., İ.Ç. and E.G.; Supervision: İ.Ç. and G.Ö.; Funding acquisition; İ.Ç. and T.U.G. All authors have read and agreed to the published version of the manuscript.

**Funding:** The authors would like to express their gratitude to the Turkish Scientific and Technological Research Council (TUBITAK) for financial support in separate projects for molecular characterization (1919B012102183) and the technological traits for some isolates (121O580).

**Institutional Review Board Statement:** Not applicable.

**Informed Consent Statement:** Not applicable.

**Data Availability Statement:** The data presented in this study are available on request from the first author. The data are not publicly available due to restrictions by the research group.

**Conflicts of Interest:** The authors declare that there is no conflict of interest.

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
