# Peer review of "Molecular and Technological Characterization of Saccharomyces cerevisiae from Sourdough"

_fermentation, doi:10.3390/fermentation9040329_

Round 1
Reviewer 1 Report
The authors propose an interesting method to investigate genetic variability of S. cerevisiae yeasts and select relevant strains to be used as starter cultures. In my opinion, major revision is required to clarify some points about the technological characterisation of isolates.
- Material and methods:
Growth assays and resistance to lactic and acetic have been evaluated with 3 CFU/mL of yeasts. Is that correct? Can you justify or explain the reason to choose the concentration of yeasts inoculated in these assays? It seems to be a low concentration of cells, which can lead to experimental errors in the concentration of inoculated cells. Please, provide the methods used to assess the concentration inoculated in the assays.
To determine fermentation rate, glass fermentation traps were filled with destilled water. This method to determine loss of CO2 during fermentation has been described previously but the special glass device (Müller valve) is filled with concentrated sulfuric acid to allow the release of CO2. Is that the device used in this determination? Can you provide a reference to justify the use of water instead of sulfuric acid?
- Results:
I suggest to improve table 5 and hence make more clear the results of technological characteristics of isolates expressing the real value ​​of each parameter analyzed with its standard error. Expressing this results as intervals (low, moderate, high…) supposes a loss of information that could be relevant.
Author Response
Dear Reviewer 1,
Thank you for going through our manuscript. The responses are attached as a PDF file.
Sincerely.

Reviewer 2 Report
The authors provided a valuable research manuscript. However, some minor observations are proposed to be taken into account.
Figure 1: Please specify in the figure title the meaning of AER, BSR, CAR
Line 118: seven or eight primers?
Line 142: please specify how many repetitions were used in the technological tests to assure the statistical data
Line 255: Could you please, give some more details on how you did you choose the 29 isolates representing each sub-cluster?
Line 336: “The polymorphic band … around 800 bp was most associated with the fermentation capacity”. Can you provide a reference in this regard?
Line361: please specify the origin of the yeast reported by Liu [18]
Line 407: could you please, specify again, the origin of the reported strains? Just to make it clear that it is or not the same food matrix at the origin
Line 425: replace Saccharomyces with S.
Author Response
Dear Review 2,
Thank you in advance for going through our manuscript. We've made your suggestions. Our responses are attached as a PDF file.
Sincerely

Round 2
Reviewer 1 Report
-